# The Role of Glucocorticoid Receptor in the Pathophysiology of Pituitary Corticotroph Adenomas

**DOI:** 10.3390/ijms23126469

**Published:** 2022-06-09

**Authors:** Daniela Regazzo, Alessandro Mondin, Carla Scaroni, Gianluca Occhi, Mattia Barbot

**Affiliations:** 1Endocrinology Unit, Department of Medicine-DIMED, University Hospital of Padova, 35128 Padova, Italy; daniela.regazzo@unipd.it (D.R.); alessandro.mondin@aopd.veneto.it (A.M.); carla.scaroni@unipd.it (C.S.); 2Department of Biology, University of Padova, 35128 Padova, Italy; gianluca.occhi@unipd.it

**Keywords:** glucocorticoid receptor, corticotroph adenomas, Cushing’s disease, HSP90, relacorilant

## Abstract

Adrenocorticotropic Hormone (ACTH)-secreting pituitary adenomas are rare tumors characterized by autonomous ACTH secretion with a consequent increase in circulating cortisol levels. The resulting clinical picture is called Cushing’s disease (CD), a severe condition burdened with high morbidity and mortality. Apart from increased cortisol levels, CD patients exhibit a partial resistance to the negative glucocorticoid (GC) feedback, which is of paramount clinical utility, as the lack of suppression after dexamethasone administration is one of the mainstays for the differential diagnosis of CD. Since the glucocorticoid receptor (GR) is the main regulator of negative feedback of the hypothalamic–pituitary–adrenal axis in normal conditions, its implication in the pathophysiology of ACTH-secreting pituitary tumors is highly plausible. In this paper, we review GR function and structure and the mechanisms of GC resistance in ACTH-secreting pituitary tumors and assess the effects of the available medical therapies targeting GR on tumor growth.

## 1. Introduction

Adrenocorticotropic hormone (ACTH)-secreting pituitary tumors are rare pituitary neoplasia characterized by autonomous, yet still responsive, ACTH secretion. Elevated ACTH levels produce cortisol excess that results in a severe clinical condition named Cushing’s disease (CD), the most common cause of endogenous hypercortisolism [1].

CD is a severe condition burdened by increased morbidity and mortality; thus, it requires prompt diagnosis and treatment [2]. Although significant improvements in surgical remission rate and available therapies have been made so far, recurrent/persistent cases are still an open issue in the management of CD [3]. In this scenario, pituitary directed drugs represent a valuable option to control hormone excess but currently available pituitary directed drugs proved satisfying results only in a reduced proportion of patients [4]. Therefore, the understanding of CD molecular background represents a key step toward the development of novel therapeutic target, which can widen the rate of treatment responders.

Lately, in addition to rare germline mutations associated with familiar CD (i.e., multiple endocrine neoplasia type 1 (MEN1) and aryl hydrocarbon receptor interacting protein (AIP) mutations), somatic mutations in novel CD predisposing genes—e.g., *ubiquitin specific peptidase 8 (USP8)* and *48 (USP48)* and *BRAF*—have been discovered in sporadic CD [5]. Mutations in these genes lead to the upregulation of the epidermal growth factor receptor (EGFR) and enhanced the promoter activity of the proopiomelanocortin (*POMC*) and *POMC*-related genes, thus supporting their involvement in the pathogenesis of CD. Despite mutations in these genes being present in a significant proportion of sporadic CD (*USP8* and *USP48* mutations account for around 50% of cases) [6], many patients still remain genetically undiagnosed. In such scenarios, the role of glucocorticoid receptor (GR) in the initiation and progression of the disease is intriguing. Indeed, despite supraphysiological cortisol levels, tumoral corticotroph cells appear, at least partially, resistant to its negative feedback [7]. This feature is crucial in clinical practice as the lack of cortisol suppression after dexamethasone challenge is one of the validated screening tests for CD [7].

Although, until recently, somatic mutations altering GR were considered only marginally involved in glucocorticoid (GC) resistance in CD, their role has come back to the limelight thanks to the advent of next-generation sequencing (NGS) that proved GR mutations are not so rare events as previously thought [6]. Defining their prevalence, however, will demand researchers’ significant efforts.

In this short overview, we examine GR function and structure, the mechanisms of GC resistance in ACTH-secreting pituitary tumors, and assess the effects of the medical therapy targeting GR on tumor growth.

## 2. GC Receptor

The human GC receptor (GR) belongs to the nuclear receptor superfamily of transcription factors (TFs). It mediates the action of GC, and it is encoded by the sole *NR3C1* gene, which contains ten exons. Different mechanisms, such as alternative transcription initiation sites, the presence of multiple possible alternative translation start sites in exon 2, and alternative splicing, generate various GC protein isoforms—mainly GRα and GRβ—from a single gene [8].

GRα is a 97 kDa protein [9], and it is constitutively and ubiquitously expressed in almost all cells, including pituitary ACTH-secreting cells, where it regulates the physiological synthesis of ACTH through a negative feedback mechanism [10]. GRβ instead, while initially considered purely an inhibitor of the transcription factor activity of GRα [11], is now recognized as having intrinsic activities in inflammatory processes, insulin signaling, cell communication, and tumorigenesis in a GRα-independent manner (see [12] for a more comprehensive review).

The overall GR structure is similar to that of other nuclear receptors. The N-terminal portion of GR contains a transactivation domain (NTD) with the ligand independent activation function 1 (AF-1) and a zinc-finger DNA-binding domain (DBD) that is also involved in GR dimerization. A hinge region follows then a C-terminal ligand-binding domain (BD) with the GC binding site and a second region with activation functions (AF2). DBD and LBD are highly conserved between different nuclear receptors [13]. The hinge region confers instead flexibility between DBD and LBD and is sensible to acetylation [14], suggesting that this process could be implicated in the regulation of GR function.

In the absence of the ligand, the inactive GRα is part of a cytoplasmic complex which includes the molecular chaperones heat shock protein (HSP) 90 and/or HSP70, p53 and the immunophilins (e.g., FK506-binding protein 51 (FKBP51) and 52 FKBP52) [15]. Upon ligand binding, GR undergoes conformational changes that triggers the release of the repressors and GR migration into the nucleus. Here, GR may exert its genomic effects by different mechanisms: the activated GR may, indeed, either directly bind positive (GRE) or negative (nGRE) regulatory elements, or GR tether to some interaction partners, mostly TFs, already bound to DNA. Alternatively, in a third model that is somewhere in between, an active GR requires both GRE/nGRE and physical interaction with other transcription factors to exert its transcriptional activity [16]. Of note, GR binds to GRE by forming either homo- or heterodimers [17]. Although not fully elucidated, the role of GR dimerization has been under debate in the GR resistance phenomenon [17]. In addition, some more recent studies demonstrate that GR could form tetramers or dimers of dimers [18]. However, the GR dimer hypothesis is still the most generally accepted model for GR mechanism of action.

The assembly of GR interactors, other transcription factors, and co-activators/co-repressor factors contribute to activating or repressing the transcription of GC-responsive genes in a cell context-specific manner. Among the transcription factor interacting with GC the nuclear factor-KB (NF-KB), the activator protein-1 (AP-1), signal transducers, and activators of transcription (STATs) are the most studied [17].

In normal corticotroph cells, the transcription of the ACTH encoding gene *POMC*, is negatively regulated by GC throughout a negative feedback mechanism, which limits the duration and the entity of GC’s effects [19].

In *POMC* regulation, GR acts through a mechanism of transrepression, thus antagonizing the binding of orphan nuclear receptors Nur77 and Nurr1 to *POMC* promoter [20,21] (Figure 1). Further studies demonstrated that three monomers of GR are recruited in protein–protein interactions with Nurr factors. This complex requires other co-regulators such as Brahma-related Gene 1 (BRG1) and the adenylpyrophosphatase (ATPase) component of the SWI/SNF chromatin remodeling complex that is constitutively on POMC promoter before GR activation [19]. Together with BRG1, Histone deacetylase 2 (HDAC2) is involved in GR complex and its recruitment is ligand dependent. GR recruitment decreases histone acetylation at the POMC promoter and gene body [22].

Other elements could be involved in the GR transrepression regulation on POMC. Instead, a 7 kb enhancer has been discovered in human POMC. This region is highly conserved and has species-specific characteristics and could be responsible for corticotropin-releasing hormone (CRH) activation and transrepression GR repression [23].

## 3. Glucocorticoid Resistance

The GC-mediated negative feedback that physiologically characterizes the HPA axis is typically harmed in ACTH-secreting pituitary adenomas, and a partial GC resistance is commonly observed [7]. In human primary corticotroph tumor cultures, cortisol administration significantly decreases ACTH secretion, suggesting a preserved GC feedback mechanism [24]. However, a similar inhibitory effect on *POMC* transcription and ACTH synthesis/secretion was not observed after dexamethasone administration to corticotroph tumor cells [25], supporting the partial nature of this feature.

Ongoing research indicates that multiple factors may contribute to both the response and the development of GC resistance in CD. All these factors may impact GR functions by reducing its availability or altering structural and/or enzymatic properties, including agonists binding [26].

Among others, somatic mutations altering the *NR3C1* genetic locus have rarely been reported in ACTH-secreting pituitary adenomas [27]. 

However, more recent studies based on exome-wide sequencing approaches identified truncating variants at the somatic levels in about 10% of cases [28,29]. Still, this frequency must be interpreted with circumspection, as similar studies failed to find comparable results [30,31,32]. A much more common molecular genetic alteration observed in the GR genetic locus is the loss of heterozygosity (LOH), which might represent a plausible explanation of the relative resistance to the inhibitory feedback of cortisol in ACTH-secreting pituitary tumors [27].

Moreover, GR downregulation and unbalanced splice variants did not seem implied in the resistance to corticosteroid feedback in ACTH-secreting pituitary adenomas. The high GRα/GRβ ratio observed in corticotropinomas was similarly present in normal pituitary [33].

Further details on GR mutations and clinical correlations in corticotropinomas will be discussed in the sections below.

Variable expression of other factors involved in mediating sensitivity to GCs, such as HSP90, TR4, BRG-1/HDAC2, and Hydroxysteroid 11-Beta Dehydrogenase 2 (HSD11B2), seems to be more relevant in determining this feature [22,34,35], Table 1.

### 3.1. HSP90

Hsp90 is a constitutively expressed molecular chaperone that facilitates the folding, correct maturation, and degradation of a wide variety of client proteins that are in turn involved in several cellular processes, including signaling transduction, DNA repair, cell cycle, and cellular differentiation [36]. HSP90, which under physiological stress conditions can represent up to 4–6% of total protein content [37], exerts its functions in concert with a number of co-chaperone proteins, such as HSF1, HSP70, p23, Hop, and Aha, forming so-called HSP90-containing protein complexes. Co-factors modulate HSP90 activity with several mechanisms—e.g., modulate ATPase activity or conformational change or targeted clients to HSP90 [38].

The ATP-dependent cycle of GR folding represents one of the most complete models of HSP90 function and of particular importance for GR function, as its affinity to ligands is strongly related to GR folding. To increase the low ligand binding affinity state, indeed, the GR conformation has to change to increase the accessibility of a hydrophobic cleft in the ligand-binding region [39]. The HSP90-containing protein complex is not only essential for the opening of this cleft [40], but it guarantees the cyclical nature of this transition [41,42] (for more details on the involvement of an HSP90 Heterocomplex on GR maturation, see [43]).

Higher levels of HSP90 have been reported in a wide variety of tumor types, thus suggesting a pivotal role in the survival and growth of malignancies. The reasons for such an increase have been ascribed to a protective effect exerted by HSP90 from various stress states, including hypoxia and ischemia, which the cells may face under pathologic conditions [36].

Higher HSP90 expression has been reported also in corticotropinomas compared to the normal pituitary gland and to non-functioning pituitary adenomas [34]. Such enhanced levels are associated with an increased binding to GR that inhibits its dissociation from the chaperone complex and its translocation to the nucleus, which otherwise suppresses *POMC* transcription and disrupts GC-mediated negative feedback [43]. 

Various HSP90 inhibitors that act by inhibiting the activity of either the C- (i.e., Silibinin) or N-terminus (i.e., CCT018159 and 17-AAG)—i.e., both domains involved in the GR maturation [34]—have been tested in corticotroph tumors models [34,44,45]. Both types of drugs suppress ACTH synthesis and secretion in corticotroph tumor cells. Moreover, CCT018159 and Silibinin ameliorated the symptoms of hypercortisolism in mice allografted with AtT20 cells [34,44]. Silibinin, with a mechanism likely related to the GR release from the HSP90 complex in a fully mature state and the consequent, increased activated GR transcriptional activity in the nucleus and reverts at least in part GC sensitivity. This further confirms the importance of HSP90 in the impairment of GC negative feedback in corticotropinoma [34].

### 3.2. TR4, Also Known as Nuclear Receptor Subfamily 2 Group C Member 2 (NR2C2)

As GR, testicular receptor 4 (TR4), also known as Nuclear Receptor Subfamily 2 Group C Member 2 (NR2C2), belongs to the nuclear receptor superfamily. Although, traditionally considered an orphan receptor, polyunsaturated fatty acids and rosiglitazone have been recognized as TR4 ligands [46]. Once activated, it regulates the expression of its target genes; the most characterized being the *ApoE* gene [46]. It is widely expressed in many cell types, with its maximum expression in the testis, prostate, ovary, cerebellum, and hippocampus [46]. In these tissues, TR4 influences many cellular processes such as spermatogenesis, glucose homeostasis and lipid metabolism [46].

Unlike normal corticotroph cells that exhibit only weak immunopositivity for TR4, ACTH-secreting tumors, and the AtT20 cell line, they have a high intranuclear expression of TR4 [47]. A potential TR4-binding site has been suggested in the POMC promoter and was further confirmed by ChiP assays in ATt20 cells. In addition, TR4 downregulation by both gene knockdown and treatments with the targeted inhibitors MEK-162, prevents corticotroph tumor proliferation and invasion. On the contrary, TR4 overexpression induced POMC transcription and ACTH secretion [47,48]. 

In addition to its role in corticotroph tumorigenesis, TR4 has also been implicated in GC resistance in these tumors. Zhang et al. showed that TR4 could dimerize with GR, thus counteracting the repression of *POMC* transcription mediated by GR itself [48]. The interaction between GR and TR4 is direct and involves GR transrepression mediated by AP1 [47,48].

### 3.3. BRG-1/HDAC2

In addition to HSP90 and TR4, other mechanisms have been implicated in GC resistance in corticotropinomas. BRG1 and HDAC2 take part in the GR complex that binds to nGRE. Indeed, HDAC2 and BRG1 contribute to chromatin remodeling, finally acting on GR’s ability to control *POMC* transcription [22,49]. HDAC2 expression has been investigated in pituitary adenomas, and it was reduced in a relevant proportion of the cohort [50]. Therefore, the loss of BRG1 or HDAC2 could induce GC resistance, as further demonstrated by Lu et al. [51].

### 3.4. HSD11B2

HSD11B2 is a microsomal enzyme complex responsible for converting cortisol in its inactive form cortisone. In this case, the GC resistance is not mediated by a direct action on GR but by regulating cortisol metabolism. A high expression of HSD11B2 has been observed in corticotroph tumors but not in normal ACTH-secreting pituitary cells, and this overexpression could determine a reduction in the intracellular cortisol and a consequent reduced negative feedback [35,52,53].

## 4. GR Mutation and Clinical Phenotype

Pituitary microadenomas, usually smaller than 6 mm, are by far the most prevalent radiological finding in CD [1]. Micro- and macroadenomas frequently display different responses to hormonal testing with the former showing higher response to CRH [54] and the latter to DDAVP with a lower suppression after dexamethasone test [55], suggesting again a role of GR resistance in the different clinical expressions of micro and macro-ACTHomas.

A different expression of GR isoforms, with decreased GRα, was identified as a potential explanation for impaired cortisol suppression after conducting high dose dexamethasone suppression tests in CD patients [56]; however, this difference was not confirmed in later studies [57,58] that found a similar GRα expression in macro- and microadenomas [58].

In addition to the extent of GRα expression, GR mutations can also determine an impaired GC sensitivity at pituitary level. A frameshift (p.S54Tfs*16) and a nonsense (p.Q302X) variant were described in two ACTH macroadenomas [28] but no peculiar features were associated with neither of them.

In 2018, Chen et al. found two cases both harboring two variants at the *NR3C1* gene (p.R714P, p.630_630del; p.Q632X, p.S512fs), but in vitro analyses of the effects of these mutations were not reported [56].

Later studies based on next generation sequencing (NGS) reported a prevalence of somatic mutations of NR3C1 around 6% of cases studied so far, suggesting they may not be rare events in ACTH-secreting pituitary adenomas [6,59].

Last year, Miao et al. found 3/49 (6.1%) GR mutations in their series of consecutive human corticotroph tumors through NGS [59]. In vitro analysis demonstrated that c.1405C > T (p.R469X) mutation produces a truncated GR protein, whereas both c.1769A > G (p.D590G) and c.2077T > G (p.Y693D) results in reduced GR expression [59].

In addition to GR mutations, other mechanisms that remain partially unknown might be involved in different GC sensitivity; among them, GR polymorphisms may alter the response to GC response and were studied in CD patients. Despite a higher prevalence of N363S polymorphism in a cohort of CD patients compared to general population, no correlations were found with clinical picture, tumor size or surgical outcome [60]. A previous study suggested LOH in the *GR* gene polymorphisms as a possible marker of DNA rearrangement during tumor initiation, but once again, data were not sufficient to associate LOH with tumor behavior [61].

Corticotroph tumor progression (CTP) leading to Nelson’s syndrome (NS) is characterized by hyperpigmentation and an increase in tumor size in patients with refractory CD submitted to bilateral adrenalectomy (BA); tumor growth can be dramatic with severe complications including death [62]. To date, no effective predictive factors are available to a priori identify patients at high risk of developing NS, and the only option is the close post-operative monitoring of all patients [62]. 

Since the marked increase in ACTH levels always precedes tumor enlargement, a possible implication of GR in the pathogenesis of NS seems at least plausible. Therefore, Karl et al. hypothesized a selective GC resistance in corticotropinoma that might contribute to ACTH secretion and accelerate tumor growth. In four cases with NS, they found a somatic frame-shift mutation in the N-terminal region, causing the premature termination of GR protein translation. However, as authors stated, the identification of a single gene defect, such as the somatic mutation of the GR should be considered a contributing factor rather than a sole causative factor of the aggressive behavior of some ACTH-secreting tumors [63].

The most severe expression of pituitary lesions are carcinomas, which fortunately are very rare. Among them, most cases are either PRL or ACTH-secreting lesions. Cerebrospinal or systemic metastases are the hallmark to diagnose pituitary carcinoma. The expression of GR mRNA was confirmed in all four carcinomas (three NS and one CD), even though not quantified [64], confirmed that even aggressive tumor maintained GC sensitivity to some extent.

The 2017 World Health Organization (WHO) classification of pituitary tumors identified three main cell lineages on the basis of specific transcriptional factors; T-Pit is typically expressed by corticotropinomas, including silent corticotroph adenomas (SCAs) [65]. SCAs are clinically non-functioning adenomas, as no signs or symptoms of cortisol excess are present, but they are immunopositive for ACTH; due to the lack of clinical features of hypercortisolism, there is frequently a delay in diagnosis with severe tumor mass-related symptoms at presentation, leading them directly to neurosurgeons [65]. They usually have a more aggressive behavior than other clinically nonfunctioning pituitary tumors. The common background with their clinically active counterpart is witnessed by the possible, although rare, transformation from SCA to CD and vice versa [66].

*GRα* mRNA steady-state levels are similar in SCAs and ACTH-secreting micro- and macroadenomas [58]. A very recent paper by Mossakowska et al. compared functioning corticotropinomas and SCAs (28 vs. 20 cases, respectively); while the authors did not find any differences in patients’ age at presentation, tumor invasiveness, proliferation index, and proportions of sparsely and densely granulated adenomas, they observed instead a higher expression of miR-124-3p in ACTH-secreting tumors, leading to GR downregulations, which in turn reduces the effect of GR feedback on corticotroph adenoma [67].

A previous paper reported higher expression of miR-200a and miR-103 in silent than in functioning corticotropinomas and higher levels of miR-488, miR-200a, and miR-103 in larger tumors irrespective of functioning status, suggesting their possible role in tumor growth [68].

## 5. Glucocorticoid Receptor Antagonists and Corticotroph Tumor Behavior

Targeting GR is a valuable option to reduce cortisol-related complications of Cushing’s syndrome (CS). Two main compounds have been developed for CS treatment, namely mifepristone and relacorilant, and the latter currently under investigation [69].

Mifepristone (Korlym^®^) is a competitive GR and progesterone receptor (PR) antagonist, and it is best known for its use as abortion pill. It was approved by the Food and Drug Administration in 2012 for the treatment of hyperglycemia associated with CS when surgery is not feasible or proved ineffective [69].

Classical hormone assessment performed during medical treatment of CS is not helpful in patients treated with mifepristone, as both ACTH and cortisol are increased due to GR antagonism. Therefore, only clinical examination and biochemical testing are used to assess treatment effectiveness. The marked increased in cortisol levels saturates the capability of 11βHSD2 to inactivate cortisol to cortisone causing a pseudo-hyperaldosteronism with consequent hypokalaemia, hypertension, and oedema. Furthermore, due to PR antagonist, endometrial thickening and vaginal bleeding are frequent in fertile women [70,71,72].

In addition to these expected adverse events, less is known about the direct effect of mifepristone on ACTH-secreting pituitary adenoma. Theoretically, by blocking the pituitary GR, the lack of cortisol negative feedback at this level should lead to adenoma growth, similarly to what happens after BA in patients developing NS [73]. 

Indeed, in a multicentre study all CD patients presented an increase in ACTH, being more than two-fold in 68.2% of cases; on the contrary, patients with ectopic hypercortisolism did not present any increase in ACTH levels. Nevertheless, mifepristone showed no significant effect on tumor size in CD patients after 6 months, except in one case where tumor growth led to treatment discontinuation after 10 weeks [74]. The extension study over a treatment period ≥ 12 months confirmed that ACTH elevation occurred in a dose-dependent manner during the treatment, as previously reported in healthy subjects [75]. ACTH levels took several weeks to achieve the plateau level and then remained stable until drug discontinuation. During treatment, no tumor modification was observed in most cases (*n* = 30). Four patients showed tumor progression; at the study’s entry, three had a macroadenoma, whereas the other had negative MRI progressing to visible lesions after 25 months. There are no sufficient data to state whether this was a direct effect of the treatment or merely a spontaneous progression of aggressive pituitary tumor that would have happened anyway. Curiously, two cases of tumor regression were also reported, a macroadenoma and a microadenoma, and the latter no longer visible after 24 weeks. An ACTH increase was similar in these three groups (stability, progression, and regression), suggesting that ACTH elevation during mifepristone therapy does not predict tumor behavior [76]. 

Relacorilant is a novel highly selective non-steroidal GR modulator, with ongoing phase III studies (NCT03697109, NCT04308590, and NCT04373265) after promising results shown in the phase II study on CS cases with glucose metabolism impairment and/or arterial hypertension. Relacorilant induces lower ACTH and cortisol increases in ACTH-dependent CS compared to mifepristone, thus preventing 11βHSD2 saturation and typical side effects of mineralocorticoid excess. On the contrary, the drug showed important benefits on blood pressure and no cases of hypokalemia were registered. Moreover, its molecular structure strongly reduces its affinity for PR avoiding anti-progesterone effect [77].

Terzolo et al. presented two cases CD who received bridging therapy with relacorilant before surgery. Both patients had a pituitary macroadenoma and experienced “unexpected” tumor shrinkage after only three months of therapy [78]. It can be speculated that this effect might depend on the increased “sensitization to endogenous somatostatin” due to the reduction in effective cortisol levels and consequent re-expression of somatostatin receptor 2 (SSTR2), previously downregulated by hypercortisolemia [79]. The effect of cortisol excess on the pituitary SSTRs pattern is well documented and SSTR2 downregulation can explain first-generation somatostatin receptors ligands (SRLs) efficacy in acromegaly but not in CD patients unless after BA [80]. The sensitization hypothesis is well supported by both in vitro and in vivo observations. Relacorilant prevented dexamethasone-induced SSTR2 downregulation in murine at-T20 cells in a dose-dependent fashion [79]. A similar observation was reported in corticotroph tumors of patients treated preoperatively with other cortisol lowering medications, but only at the mRNA level [81]. Recently, two patients with ectopic ACTH secretion receiving relacorilant showed increased tumor uptake of the tracer at somatostatin-based nuclear imaging that indirectly confirms the re-expression of biologically active SSTR2. Moreover, one of them showed the restoration of pituitary gland signals at repeated scans [79].

Further studies are needed to confirm these findings and to potentially introduce new treatment approaches that combine SRLs to GR antagonists.

## 6. Steroidogenesis Inhibitors and Corticotroph Tumor Behavior

Although steroidogenesis inhibitors do not directly target GR, the reduction in circulating cortisol levels can cause a loss of negative feedback at the pituitary level, thus promoting tumor growth with a mechanism similar to that of BA. However, given the scant data available, whether this mechanism promotes adenoma growth in CD is still to be elucidated. A real-life study on patients treated with metyrapone did not show tumor enlargement over a mean treatment period of 9 months [82]. A decrease in tumor volume was instead reported in a patient potentially harboring GC-mediated positive feedback on the adenoma [83]. Vice versa, the visualization of pituitary adenoma in a previously negative MRI has been reported for both ketoconazole [84] and mitotane therapy [85] in a few cases (Table 2). A phase III study on osilodrostat reported variable effects on tumor size, with patients exhibiting growth (33%), shrinkage (38%), or stability in tumor size over 48 weeks of treatment irrespective of baseline tumor volume or ACTH levels at baseline. However, no newly visible adenoma was diagnosed in patients with negative MRI at baseline [86]. Still, it should be mentioned that several patients presenting tumor shrinkage had previous pituitary irradiation that might be responsible itself for tumor regression. Interestingly, a recent case report described a late-onset progression during treatment [87]. Although data on tumor progression while on steroidogenesis inhibitors are conflicting, periodic monitoring of pituitary imaging should be recommended during treatment [88].

## 7. Conclusions and Future Remarks

Although the GR is often seen in the background of the pathogenesis of CD, the prevalence and significance of its alterations in the initiation and progression of CD are still under debate. As we have shown in this review, NGS technologies turned the spotlight back on the role of GR mutations on the impaired GC sensitivity at the pituitary level in CD patients. The number of patients analyzed so far, however, is still too low to define the real prevalence and impact of GR mutations on CD pathogenesis. On the other side, however, they are surely less prevalent than those found in the deubiquitinases USP8 and USP48, thus suggesting a more likely contribution in the modulation of tumor phenotype rather than in the transformation and/or progression. Moreover, we have highlighted the importance of other components of the GR complex, such as HSP90, in the impairment of GC signaling and sensitivity at the pituitary level, but possible additional functions of GR that have never been evaluated so far in CD, including those related to its mitochondrial localization as well GC-dependent chromatin decompaction, may represent something to look at in the next few years. 

Last but not least, the unpredictable effect on tumor volume due to cortisol blockade confirms that the understanding of the role of GR complex in ACTH-secreting tumors is still defective and may involve other signaling pathways. Among others, SSTRs signaling might have a prominent role in this action, as they are highly influenced by circulating GC levels and can positively impact tumor volume. Furthermore, there is increasing evidence that GCs, including endogenous cortisol, may have a leading role in solid tumors progression and chemoresistance; hence, it might have a direct, but still undetermined, role in corticotroph adenoma growth as well.

Current uncertainty imposes a periodic ACTH and radiological monitoring for patients on steroidogenesis inhibitors or GC receptor antagonists to early detect tumor growth. Ongoing and future studies will fill the knowledge gap about the long-term effect and impact of these compounds on tumor growth and the potential for more effective treatment combinations. 

## Figures and Tables

**Figure 1 ijms-23-06469-f001:**
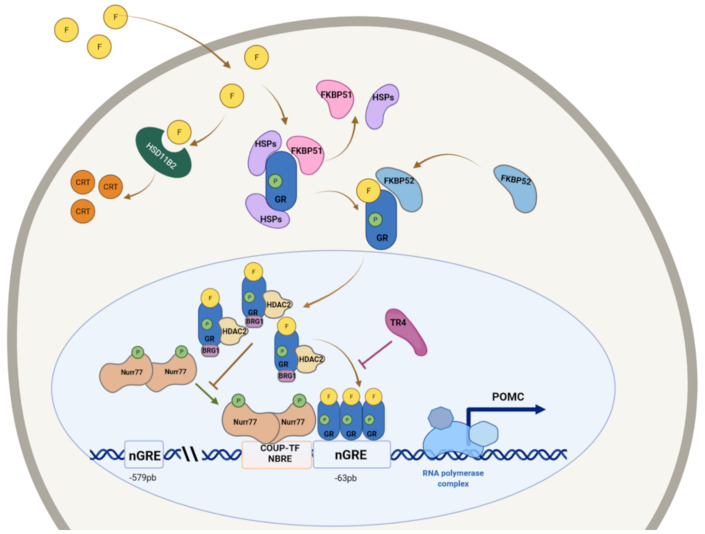
Graphical representation of glucocorticoid receptor (GR) action in ACTH-secreting pituitary cells and regulation of POMC expression. GR resides in the cytoplasm inactive in a multimeric complex with heat shock proteins (HSP90) and immunophilins (FKBP51 and/or FKBP52). After cortisol binding, GR dissociates from HSP90, and nuclear localization signal activates. Associated with cortisol (F), GR translocates into the nucleus and combines with other two COR/GR complexes, binding to nGRE (negative Glucocorticoid Receptor Element) in the POMC promoter. GR downregulates POMC expression not by directly binding nGRE but via transrepression, antagonizing the binding of orphan nuclear receptors Nur77 and Nurr1. This mechanism also requires the recruitment of BRG1 protein and Histone deacetylase 2 (HDAC2), decreasing histone acetylation of the POMC gene and, consequently, its expression. NBRE: Nur77-binding response element; P: phosphorylated; other factors possibly involved in GC resistance: TR4 (Nuclear Receptor Subfamily 2 Group C Member 2) and HSD11B2 (Hydroxysteroid 11-Beta Dehydrogenase 2), which converts F to the inactive cortisone (CRT).

**Table 1 ijms-23-06469-t001:** Schematic summary of the main factors that might influence GR action, their physiological role, and the mechanisms by which they impaired GC sensitivity at pituitary levels.

Gene/Protein	Role	Type of Impairment	Mechanism of Action
Somatic *NR3C1* mutation	To determine GR expression	Gene inactivation, GR resistance to negative feedback to GC	Impaired GR functioning
Somatic *NR3C1* Loss of Heterozygosity	To determines GR expression	Gene inactivation, GR resistance to negative feedback to GC	Reduced expression of functional GR
HSP90	Molecular cochaperone in GR complex	Higher expression compared to normal pituitary	Increased binding to GR inhibiting its dissociation from GR complex and translocation to the nucleus
TR4	Orphan nuclear receptor	Higher intranuclear expression compared to normal pituitary ACTH-secreting cells	Dimerization with GR counteracting GR repression on POMC transcription
BRG1/HDAC2	Molecular interactors, part of GR complex	Lower expression compared to normal pituitary	Loss of BRG1 or HDAC2 could induce GC resistance
HSD11B2	Microsomal enzyme converting cortisol to inactive cortisone	Higher expression compared to normal corticotroph cells	Reduction of intracellular cortisol with consequent reduce negative feedback on GR

**Table 2 ijms-23-06469-t002:** Effects of steroidogenesis inhibitors on corticotropinomas’ volume.

Drug	Mechanism of Action	Study Design	Duration	Effect on Tumor Volume	Cases	Study
Ketoconazole	Side-chain cleavage 17, 20 lyase 11β-hydroxylase Aldosterone synthase	Retrospective	na	Visualization of the adenoma after a previously negative MRI	8/58 *	Castinetti F, 2014 [84]
Mitotane	Side-chain cleavage 11β-hydroxylase	Retrospective	10, 9 months **	Visualization of the adenoma after a previously negative MRI	12/48 *	Baudry C, 2012 [85]
Metyrapone	11β-hydroxylase Aldosterone synthase	Case report	2 months	Tumor shrinkage	1	Tsujimoto Y, 2021 [83]
Osilodrostat	11β-hydroxylase Aldosterone synthase	Prospective	48 weeks	Tumor volume decrease ≥20%	24/64	Pivonello R, 2020 [86]
				Tumor volume increase ≥20%	21/64	
		Case report	4 years **	Late onset tumor increase	1	Fontaine-Sylvestre C, 2021 [87]

* Proportion of patients with baseline negative MRI presenting newly visible adenoma; ** mean time to effect; na = not assessed, MRI = Magnetic resonance imaging.

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
