# Peer review of "The Role of Glucocorticoid Receptor in the Pathophysiology of Pituitary Corticotroph Adenomas"

_ijms, 2022, doi:10.3390/ijms23126469_

Round 1

Reviewer 1 Report

Daniela Regazzo and colleagues describes the role of glucocorticoid receptor in the pathophysiology of pituitary corticotroph adenomas. The authors presented in this manuscript is well rationalized, executed and interpreted.

Following are the suggestions to the authors that may improve the impact of this manuscript.

  1. Line 358 on page 8, Author’s stated that ‘the sensitization hypothesis is well supported by both in vitro and in vivo observations’. Being a review article could the authors elaborate on these findings, it will be good if the data should describe here so the readers can benefit more on this review.
  2. In Table 1, please give the individual reference numbers on the right side of the table for each drug.
  1. Discussion and conclusion are missing, please write discussion and conclusion separately, how it is benefitted to the patients and finally conclusion about the review.

4. The review article length is too small, if the authors address the third comment we can address.

  1. A summary table of significant effects would be useful for the readers as many different outcomes are presented in the results.

Author Response

Daniela Regazzo and colleagues describes the role of glucocorticoid receptor in the pathophysiology of pituitary corticotroph adenomas. The authors presented in this manuscript is well rationalized, executed and interpreted.

Thank you for your comment.

Following are the suggestions to the authors that may improve the impact of this manuscript.

  1. Line 358 on page 8, Author’s stated that ‘the sensitization hypothesis is well supported by both in vitro and in vivo observations’. Being a review article could the authors elaborate on these findings, it will be good if the data should describe here so the readers can benefit more on this review.

Thank you for the suggestion. We provide further information on this specific point. "The sensitization hypothesis is well supported by both in vitro and in vivo observations. Relacorilant prevented dexamethasone-induced SSTR2 downregulation in murine at-T20 cells in a dose-dependent fashion [79]. A similar observation was reported in corticotroph tumors of patients treated preoperatively with other cortisol lowering medications, but only at mRNA level [81]. Recently, Two patients with ectopic ACTH secretion receiving relacorilant showed increased tumor uptake of the tracer at somatostatin-based nuclear imaging that indirectly confirms the re-expression of biologically active SSTR2. Moreover, one of them showed restoration of pituitary gland signal at repeated scans [79]

  1. In Table 1, please give the individual reference numbers on the right side of the table for each drug.

We modified table 1 as requested.

Discussion and conclusion are missing, please write discussion and conclusion separately, how it is benefitted to the patients and finally conclusion about the review.

Thank you for the comment. We added the conclusion section to sum up the information reported in the paper.

Conclusions

“GR mutations in CD pathogenesis have again been put forward thanks to the use of NGS technology that proved they are present in about 6% of the ACTH-secreting tumors analized so far. Still, the number of cases analyzed is quite low to determined the real prevalence of these mutations compared to the most prevalent onces involving the deubquitinases USP8 and USP48. Although GR mutations are directly involved in the regulation of ACTH secretion, they probably represent a contributing rather than the causative event to tumor progression. Besides, other components of GR complex, such as TR4, HSP90 and HSP70 might be involved in impaired GC signaling and sensitivity at pituitary level, favouring corticotroph tumor growth, but their role needs to be further investigated. Moreover, the unpredictable effect on tumor volume due to cortisol blockade confirms that the understanding of the role of GR complex in ACTH-secreting tumors is still defective and imposes a periodic ACTH and radiological monitoring to early detect tumor growth. Ongoing and future studies will further clarified the long term effect and the impact of steroidogenesis inhibitors and GR antagonists on tumor growth”.

Reviewer 2 Report

With the importance of pituitary corticotroph adenomas, Regazzo et al evaluated the role of glucocorticoid receptor in the pathophysiology of this adenomas. Overall, the whole structure of this study is good. However, some corrections are recommended for providing clear information. Particularly, I listed the following comments in detail here.

Major concerns:

In the introduction, the sentences are un-regularly dispersed. Additionally, all of the names and terms should be completely mentioned for the first time in the abstract and text. For example, ACTH, MEN1, AIP, USP8, USP48, BRAF, and so on.

The references are missed for some sentences. For example, “CD is a severe condition burdened by increased morbidity and mortality, thus it requires prompt diagnosis and treatment t. Although significant improvements in surgical remission rate and available therapy have been made so far, recurrent/persistent cases are still an open issue in the management of CD; therefore, the understanding of its molecular background of CD represents a key step toward the development of novel therapeutic target. Lately, besides rare germline mutations associated with familiar CD (i.e. MEN1, AIP), somatic mutations in novel CD predisposing genes – e.g. USP8, USP48, and BRAF– have been discovered in sporadic CD. Mutations in these genes lead to upregulation of the epidermal growth factor receptor (EGFR) and enhanced promoter activity of the proopiomelanocortin (POMC) and POMC-related genes, thus supporting their involvement in the pathogenesis of CD.”, and so on.

The size of Figure 1 needs to be maximized with resolution.

In the end, what is your conclusion? Hence, add a significant statement that must be structured as “what was offered by authors? Do the authors have more thoughts on this field?

Author Response

With the importance of pituitary corticotroph adenomas, Regazzo et al evaluated the role of glucocorticoid receptor in the pathophysiology of this adenomas. Overall, the whole structure of this study is good. However, some corrections are recommended for providing clear information. Particularly, I listed the following comments in detail here.

Thank you for your comment.

Major concerns:

  • In the introduction, the sentences are un-regularly dispersed. Additionally, all of the names and terms should be completely mentioned for the first time in the abstract and text. For example, ACTH, MEN1, AIP, USP8, USP48, BRAF, and so on.

Thank you for the comment. We tried to smoothen the introduction section and we hope it has improved. All abbreviations were spelt out at their first appearance in the text as requested.

  • The references are missed for some sentences. For example, “CD is a severe condition burdened by increased morbidity and mortality, thus it requires prompt diagnosis and treatment t. Although significant improvements in surgical remission rate and available therapy have been made so far, recurrent/persistent cases are still an open issue in the management of CD; therefore, the understanding of its molecular background of CD represents a key step toward the development of novel therapeutic target. Lately, besides rare germline mutations associated with familiar CD (i.e. MEN1, AIP), somatic mutations in novel CD predisposing genes – e.g. USP8, USP48, and BRAF– have been discovered in sporadic CD. Mutations in these genes lead to upregulation of the epidermal growth factor receptor (EGFR) and enhanced promoter activity of the proopiomelanocortin (POMC) and POMC-related genes, thus supporting their involvement in the pathogenesis of CD.”, and so on.

Thank you for the observation. Further citations were added where appropriate.

  • The size of Figure 1 needs to be maximized with resolution.

We improved figure quality as requested. Please note that the high resolution image has been uploaded separately, so it is not the one that appears in the main text.

  • In the end, what is your conclusion? Hence, add a significant statement that must be structured as “what was offered by authors? Do the authors have more thoughts on this field?

Thank you for the suggestion. We added the conclusion section as suggested.

Conclusions

“GR mutations in CD pathogenesis have again been put forward thanks to the use of NGS technology that proved they are present in about 6% of the ACTH-secreting tumors analized so far. Still, the number of cases analyzed is quite low to determined the real prevalence of these mutations compared to the most prevalent ones involving the deubquitinases USP8 and USP48. Although GR mutations are directly involved in the regulation of ACTH secretion, they probably represent a contributing rather than the causative event to tumor progression. Besides, other components of GR complex, such as TR4, HSP90 and HSP70 might be involved in impaired GC signaling and sensitivity at pituitary level, favouring corticotroph tumor growth, but their role needs to be further investigated. Moreover, the unpredictable effect on tumor volume due to cortisol blockade confirms that the understanding of the role of GR complex in ACTH-secreting tumors is still defective and imposes a periodic ACTH and radiological monitoring to early detect tumor growth. Ongoing and future studies will further clarified the long term effect and the impact of steroidogenesis inhibitors and GR antagonists on tumor growth”.

Round 2

Reviewer 1 Report

As per the suggestion on the last review still one point need to be addressed.

Please write some discussion on this topic.

Author Response

Thank you for you commentary. We tried to satisfied your request, still without exceeded paper length asked by the paper policy by far. The findings presented in the text have been briefly discussed. Please consider also that this is a short review not properly an Opinion article